# Production of Recombinant Single-Chain Eel Luteinizing Hormone and Follicle-Stimulating Hormone Analogs in Chinese Hamster Ovary Suspension Cell Culture

Munkhzaya Byambaragchaa [1], Sang-Gwon Kim [2], Sei Hyun Park [2], Min Gyu Shin [3], Shin-Kwon Kim [3], Myung-Hwa Kang [4] and Kwan-Sik Min [1,2,5,*]

[1] Institute of Genetic Engineering, Hankyong National University, Anseong 17579, Republic of Korea; munkhzaya_b@yahoo.com

[2] Graduate School of Animal Biosciences, Hankyong National University, Anseong 17579, Republic of Korea; tom0391@naver.com (S.-G.K.); mrtree119@naver.com (S.H.P.)

[3] Aquaculture Research Division, National Institute of Fisheries Science, Busan 46083, Republic of Korea; smg159@korea.kr (M.G.S.); ksk4116@korea.com (S.-K.K.)

[4] Department of Food Science and Nutrition, Hoseo University, Asan 31499, Republic of Korea; mhkang@hoseo.edu

[5] Division of Animal BioScience, School of Animal Life Convergence Sciences, Hankyong National University, Anseong 17579, Republic of Korea

* Correspondence: ksmin@hknu.ac.kr; Tel.: +82-31-670-5421

**Abstract:** We produced rec-single chain eel luteinizing (rec-eel LH) and follicle-stimulating (rec- eel FSH) hormones displaying high biological activity in Chinese hamster ovary suspension (CHO-S) cells. We constructed several mutants, in which a linker, including an O-linked glycosylated carboxyl-terminal peptide (CTP) of an equine chorionic gonadotropin (eCG) β-subunit, was attached between the β- and α-subunit (LH-M and FSH-M) or in the N-terminal (C-LH and C-FSH) or C-terminal (LH-C and FSH-C) regions. The plasmids were transfected into CHO-S cells, and culture supernatants were collected. The secretion of mutants from the CHO-S cells was faster than that of eel LHβ/α-wt and FSHβ/α-wt proteins. The molecular weight of eel LHβ/α-wt and eel FSHβ/α-wt was 32–34 and 34–36 kDa, respectively, and that of LH-M and FSH-M was 40–43 and 42–45 kDa, respectively. Peptide-N-glycanase F-treatment markedly decreased the molecular weight by approximately 8–10 kDa. The $EC_{50}$ value and the maximal responsiveness of the eel LH-M and eel FSH-M increased compared with the wild-type proteins. These results show that the CTP region plays a pivotal role in early secretion and signal transduction. We suggest that novel rec-eel LH and FSH proteins, exhibiting potent activity, could be produced in large quantities using a stable CHO cell system.

**Keywords:** *eel* LH; *eel* FSH; single-chain; analog; CHO-S cell

## 1. Introduction

Luteinizing (LH) and follicle-stimulating (FSH) hormones are glycoproteins secreted from the pituitary gland, which control gonadal functions in mammals and fish [1]. These glycoproteins consist of noncovalently linked α- and β- subunits, which are composed of 92–96 and 115–121 amino acids, respectively [2]. The α- and β-subunits contain 1–2 carbohydrate-binding sites and are post-translationally modified; differences in the bound oligosaccharides confer distinctive half-lives and biological activities to these glycoproteins [3].

The family of glycoprotein hormone receptors, which includes lutropin/choriogonadotropin hormone receptor (LH/CGR) and follicle-stimulating hormone receptor (FSHR), belongs to the superfamily of G protein-coupled receptors (GPCRs) that couple extracellular agonists to intracellular effector molecules through the reactions of heterotrimeric G proteins [4–7], GPCR kinases (GRK) [8,9], β-arrestins [10–12], and extracellular signal-regulated kinases (ERK) [13,14].

Carbohydrate residues attached to the glycoprotein hormones, including LH, FSH, thyroid-stimulating hormone (TSH), and chorionic gonadotropin (CG), play a pivotal role in the biological activity of these hormones [3,15–20]. The glycosylation of these glycoproteins is generally important in determining their serum half-life and biological activity [3]. The specific glycosylation sites are essential for the signal transduction through these receptors [1]. We previously demonstrated that the oligosaccharide chains in the eel recombinant (rec) LH and FSH are very important in the signal transduction through their receptor [1,2]. However, the quantity of rec-hormones produced in Chinese hamster ovary-K1 (CHO-K1) cells for undertaking such studies is very limited. To overcome this limitation, we also produced rec-eel LH and FSH in the *Bombyx mori* (silkworm) system. Although this system produces large quantities of rec-proteins, their biological activity is not good [21].

The most common method for induction of maturation in female eels is a weekly injection of a salmon pituitary extract (SPE) and carp pituitary extract (CPE). However, these hormones extracted from freeze-dried pituitaries of salmon and carp do not provide a uniform dosage [22–24]. The artificial induction of gonadal maturation remains unclear because of the lack of suitable amounts of eel gonadotropin hormones (GTHs). Expression of potent rec-hormones in suspension cells is a feasible option. The eCG, secreted from the placenta during early pregnancy, is a unique molecule that displays both LH- and FSH-like activities in non-equid species [2]. Therefore, we chose CHO suspension (CHO-S) cells to express rec-single chain eel LH and FSH hormones containing equine CG (eCG) carboxyl-terminal peptide (eCTP) with 12 potential O-linked glycosylation sites as a linker [3].

This study was aimed at production of rec-gonadotropin with higher biological activity than that obtained in previous studies on the production of single chain eel LH and eel FSH hormones. We demonstrate the production of novel rec-eel LH and FSH analog hormones, with more potent biological activity, in mammalian cells. We suggest that these novel rec-eel LH and FSH proteins can be produced in large quantities using the stable CHO cell system.

## 2. Materials and Methods

### 2.1. Materials

The oligonucleotides used in this study were synthesized by Genotech (Daejeon, Republic of Korea). The cloning vector (pGEMT-easy) was purchased from Promega (Madison, WI, USA). The pcDNA3.1 expression vector, Freestyle MAX reagent, FreeStyle CHO-suspension (CHO-S) cells, Lipofectamine-2000, and Assay Complete medium were purchased from Invitrogen Corporation (Carlsbad, CA, USA). The pCORON1000 SP VSV-G-tag expression vector was purchased from Amersham Biosciences (Piscataway, NJ, USA). Ham's F-12 medium, fetal bovine serum (FBS), OptiMEM medium, and CHO-S-SFMII medium were purchased from Gibco BRL (Grand Island, NY, USA). Monoclonal antibodies for Western blotting were produced in our laboratory [25,26]. The rec-eel LH and FSH ELISA systems were developed in our laboratory [1,26]. Restriction enzymes and DNA ligation and polymerase chain reaction (PCR) reagents were purchased from Takara Bio (Shiga, Japan). FBS was obtained from HyClone Laboratories (Logan, UT, USA). The deglycosylation kit (PNGase) was purchased from New England Biolabs (Ipswich, MA, USA). The cAMP Dynamic 2 immunoassay kit was purchased from Cisbio (Codolet, France). CHO-K1 cells were obtained from the Korean Cell Line Bank (KCLB, Seoul, Republic of Korea). The QIAprep-Spin plasmid kit was purchased from Qiagen Inc. (Hilden, Germany), and disposable spinner flasks were from Corning Inc. (Corning, NY, USA). Centriplus Centrifugal Filter Devices were purchased from Amicon Bio separations (Billerica, MA, USA). All other reagents were obtained from Sigma–Aldrich (St. Louis, MO, USA).

### 2.2. Construction of Vectors for the Expression of Eel LH-wt, FSH-wt, and Mutant Proteins

The cDNAs for single chain eel LH and eel FSH were cloned using cDNA prepared from the eel pituitary, as reported previously [1]. We constructed several LH and FSH

mutants, with the C-terminal region of the eCG β-subunit (eCTP: 115–149 amino acids) inserted between the β- and α-subunit (LH-M and FSH-M), or in the N-terminal (C-LH and C-FSH) or C-terminal (LH-C and FSH-C) regions, using PCR, as reported previously [3].

The full-length PCR products were eluted and cloned into the pGEMTeasy vector. The clones were used to transform *Escherichia coli* DH5α competent cells. The plasmids were isolated and sequenced to check for errors introduced during the PCR. The full-length fragments of the mutants were ligated into the pcDNA3 mammalian expression vector at the EcoRI and XhoI sites. A schematic of each mutant, in which the eCTP is attached between the β- and α-subunit or in the N-terminal region of the β-subunit, is presented in Figure 1. The plasmids (designated as pcDNA3-eel LHβ/α-wt, LHβ/α-M, C-LHβ/α, FSHβ/α-wt, FSHβ/α-M, and C-FSHβ/α) were then purified and sequenced from both the ends using automated DNA sequencing to ensure the veracity of the mutations.

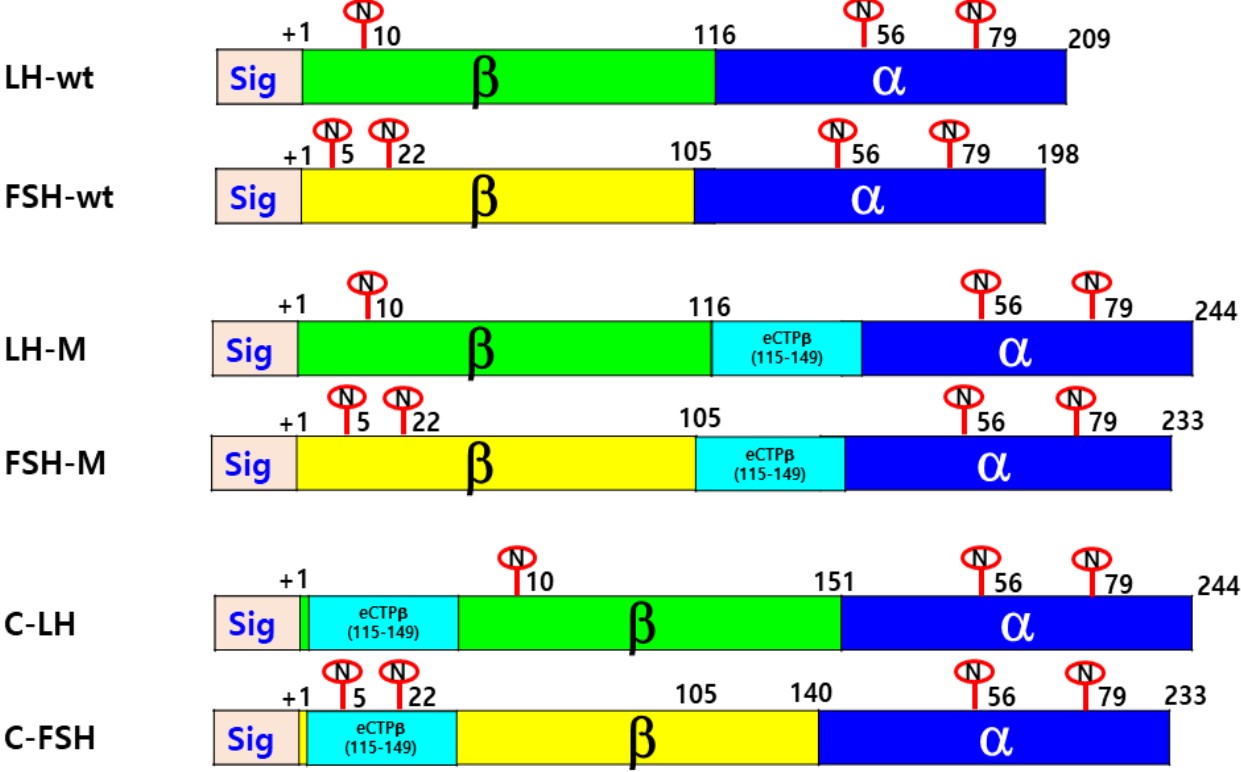

eCTPβ (115-149): **SSSS**KD PP**S**QPL**TSTS** TPTPGA**SRRS** SHPLPIK**TS**

**Figure 1.** Schematic diagrams of rec-eel LHβ/α-wt, eel FSHβ/α-wt, and mutants. The single-chain forms (wt) of eel LHβ/α and eel FSHβ/α were engineered to contain the β-subunit and common α-subunit sequences, as previously described [1,2]. In the mutants, the carboxyl-terminal peptide (CTP) region of O-linked oligosaccharides in the eCG β-subunit was inserted between β- and α-subunit or at the N-terminal region of the β-subunit using PCR. This 35-amino acid linker encodes the CTP of the equine CG β-subunit. The six expression vectors (designated as pcDNA3-eel LHβ/α-wt, LH-M, C-LH, FSHβ/α-wt, FSH-M, and C-FSH) encoded eel LHβ/α-wt, FSHβ/α-wt, and their respective mutants. The eCTP linker in the C-LH and C-FSH mutants was inserted between the first and second amino acid residues of the β-subunit of the mature eel LH and eel FSH proteins. The numbers indicate the amino acids in the mature proteins. The encircled "N" denotes N-linked oligosaccharide on the eel LH, eel FSH, and mutants, respectively. The numbers in addition to the encircled "N" indicate the N-linked glycosylation sites. The green color is LHβ, the yellow color is FSHβ, α-subunit is blue color, and the skyblue color is the eCTPβ. The red colors in the eCTPβ (115–149) denote potential O-linked oligosaccharide sites.

## 2.3. Expression of Rec-Eel LH-wt, Rec-Eel FSH, and Mutant Proteins

For the expression of rec-proteins, the expression vectors were transfected into CHO-S cells using the FreeStyle$^{TM}$ MAX reagent, as described previously [7]. The CHO-S cells were cultured in FreeStyle CHO expression medium at $1 \times 10^7$ cells per 30 mL of medium for 3 days. One day prior to transfection, CHO-S cells were passaged at $5 \times 10^5$ cells/mL. Flasks were placed on an orbital shaker platform rotating at 120–135 rpm at 37 °C in a humidified atmosphere of 8% $CO_2$ in air. On the day of transfection, the cell density was approximately $1.2–1.5 \times 10^6$ cells/mL. The plasmid DNA was diluted into the OptiPRO$^{TM}$ serum-free medium (SFM) and FreeStyle$^{TM}$ MAX reagent. Thereafter, the samples were mixed gently by inversion and incubated for 10 min at RT to allow the formation of complexes. The DNA-FreeStyle$^{TM}$ MAX reagent complex was slowly added to the cells. The transfected cells were cultured at 37 °C in a humidified atmosphere of 8% $CO_2$ on an orbital shaker platform rotating at 135 rpm. After transfection, 2 mL culture medium was collected on days 0, 1, 3, 5, and 7 to quantify the secreted rec-proteins. Finally, the culture medium was collected on day 7 post-transfection and centrifuged at $100,000 \times g$ for 10 min at 4 °C to remove cell debris. The supernatant was collected and concentrated using a Centricon filter. The rec-proteins were diluted as required for Western blotting and enzyme-linked immunosorbent assay (ELISA).

## 2.4. Quantitation of Rec-Eel LH and Eel FSH Proteins Using ELISA

The rec-eel LH-wt, FSH-wt, and mutant proteins in the cell-culture medium were quantified via a double-sandwich ELISA using plates coated with monoclonal antibody eel FSH 5A5 (binds to the α-subunit of eel LH) and FSH 5A14 (binds to the β-subunit of eel FSH), as described previously in reports from our laboratory [27]. The wells were blocked by incubation with 1% skim milk in phosphate-buffered saline (PBS) for 1 h at 37 °C. After removal of the blocking reagent, the wells were washed with filtered PBS containing 0.05% Tween 20 (PBS-T). Next, 100 μL of rec-hormone samples were added to the wells and incubated for 1–2 h at 37 °C. After washing with PBS-T three times, 400-fold diluted horseradish peroxidase-conjugated anti eel FSH 5A11 (binds to the α-subunit of eel LH and FSH) antibody in PBS was added and the plate was incubated for 1 h at room temperature. The wells were washed five times and incubated with 100 μL of substrate solution (tetramethylbenzidine) for 20 min at room temperature. The reaction was stopped by adding 50 μL of 1 M $H_2SO_4$. The absorbance at 450 nm was measured with a microplate reader (Cytation 3, Biotek, Winooski, VT, USA). For this assay, purified rec-eel FSHβ/α protein (0–400 ng/mL) from *E. coli*, previously produced in our laboratory, was used as a standard.

## 2.5. Western Blotting Analysis of Rec-Proteins

For Western blot analysis, the concentrated sample of proteins (15 μg) from the medium was subjected to sodium dodecyl sulfate-polyacrylamide gel electrophoresis (SDS-PAGE) on a 12% reducing gel, following the Laemmli method [28]. The electrophoresed proteins were transferred onto a polyvinylidene difluoride (PVDF) membrane (0.2 μm) using a Bio-Rad Mini Trans-Blot electrophoresis cell (Hercules, CA, USA). The membrane was blocked by incubation in 5% skim milk in TBS-T (20 mM Tris-HCl, pH 7.6, 140 mM NaCl, 0.1% Tween 20) and then incubated with monoclonal anti-eel FSHα antibody for detection of rec-eel LH (designated as 5A5) and with anti-eel FSH β antibody for detection of rec-eel FSH (designated as 5A14) for 13–15 h. The blot was washed with TBS-T and incubated with horseradish peroxidase-conjugated anti-mouse secondary antibody for 2 h. The membranes were washed again, and detection was performed using an enhanced chemiluminescence system.

## 2.6. Enzymatic Release of N-Linked Oligosaccharides

To remove all *N*-linked glycans, the rec-eel LH-wt, rec-eel FSH-wt, and mutant proteins (15 μg) were incubated with PNGase F (1 μL enzyme (2.5 U/mL)/20 μL sample + 2 μL of

10X Glycobuffer + 2 μL of 10% NP-40) for 1 h at 37 °C after boiling at 100 °C for 10 min with 1 μL of 10X Glycoprotein Denaturing Buffer. The samples were subjected to SDS-PAGE and analyzed using Western blotting.

### 2.7. Analysis of cAMP Levels Using Homogenous Time-Resolved Fluorescence Assay

The eel LH/CGR and eel FSHR were subcloned into the pVSVG expression vector at XhoI and EcoRI sites and the direction of cloning was confirmed by restriction mapping and verified by sequencing the entire open reading frame. CHO-K1 cells were cultured in a growth medium (Ham's F-12 medium containing antibiotics (penicillin and streptomycin), glutamine (2 mM), and 10% FBS). These cells were transiently transfected according to the supplier's protocol, as described previously [8]. The CHO-K1 cells were grown to 80–90% confluence in six-well plates. The diluted plasmid DNAs were transfected using the Lipofectamine reagent. After 5 h, growth medium containing 20% FBS was added to each well. The transfected cells were adjusted for cAMP analysis, 48–72 h post-transfection.

The cAMP accumulation in the CHO-K1 cells expressing eel LHR and eel FSHR was quantified using the cAMP Dynamics 2 competitive immunoassay kit, as described previously [11]. The cAMP response assay used a cryptate-conjugated anti-cAMP monoclonal antibody and a d2-labeled cAMP reagent. The transfected cells (10,000 cells per well) were seeded in a 384-well plate. Compound medium buffer containing the ligand (5 μL) was added to each well and the plate was incubated for 30 min. Subsequently, cAMP-d2 and anti-cAMP-cryptate were added to each well and the plate was incubated for 1 h at room temperature. The compatible homogeneous time-resolved fluorescence (HTRF) energy transfer (665 nm/620 nm) was measured using a TriStar2 S LB942 microplate reader (BERTHOLD Tech., Wildbad, Germany). The results are represented as Delta F% (cAMP inhibition), which was calculated using the following equation: Delta F% = (standard or sample ratio-sample negative) $\times$ 100/ratio negative. The cAMP concentration for Delta F% values was calculated using the GraphPad Prism software (version 6.0; GraphPad Software Inc., La Jolla, CA, USA).

### 2.8. Data Analysis

The data for the concentration–response curves were generated from experiments performed in duplicates. GraphPad Prism 6.0 (San Diego, CA, USA) was used for analyzing the cAMP response, $EC_{50}$ values, and the stimulation curve analyses. The curves fitted in a single experiment were normalized to the background signal measured for mock-transfected cells. The results are expressed as mean $\pm$ standard error of mean from three independent experiments. GraphPad Prism 6.0 (San Diego, CA, USA) was used to evaluate the differences between samples using one-way analysis of variance, followed by Turkey's multiple comparison tests. A *p*-value of <0.05 was taken to indicate significance between the groups.

## 3. Results

### 3.1. Secretion of Rec-Eel LHβ/α-wt, Eel FSHβ/α-wt, and Mutant Proteins

The quantitation of the secretion of rec-hormones was performed by ELISA, as shown in the Materials and Methods. The mutants were constructed to investigate the functional effects of O-linked glycosylation sites of the eCG β-subunit CTP region on the biological activity of rec-eel LH and FSH hormones. First, we designed several mutants (FSH-M, LH-M, C-FSH, and C-LH) including the eel LHβ/α-wt and FSHβ/α-wt, as shown in Figure 1. To quantitate the number of proteins secreted in the cell culture medium, the expression vectors were transiently transfected into CHO-S cells.

The concentrations of rec-eel FSH-wt and eel LH-wt proteins were gradually increased until the final supernatant collection post-transfection. The expression levels of the rec-eel FSH-wt and eel LH-wt were approximately 565 $\pm$ 41 and 598.5 $\pm$ 42 ng/mL at 7 days after transfection, respectively (Figure 2). The expression levels of the eCTP containing mutants (LH-M, FSH-M, C-LH, and C-FSH) peaked at 660 $\pm$ 18.1, 433.4 $\pm$ 14.4, 570 $\pm$ 14.2,

and $355 \pm 18$ ng/mL at 1-day post-transfection. The quantity of secreted proteins was consistently high until 7 days post-transfection for all the mutants.

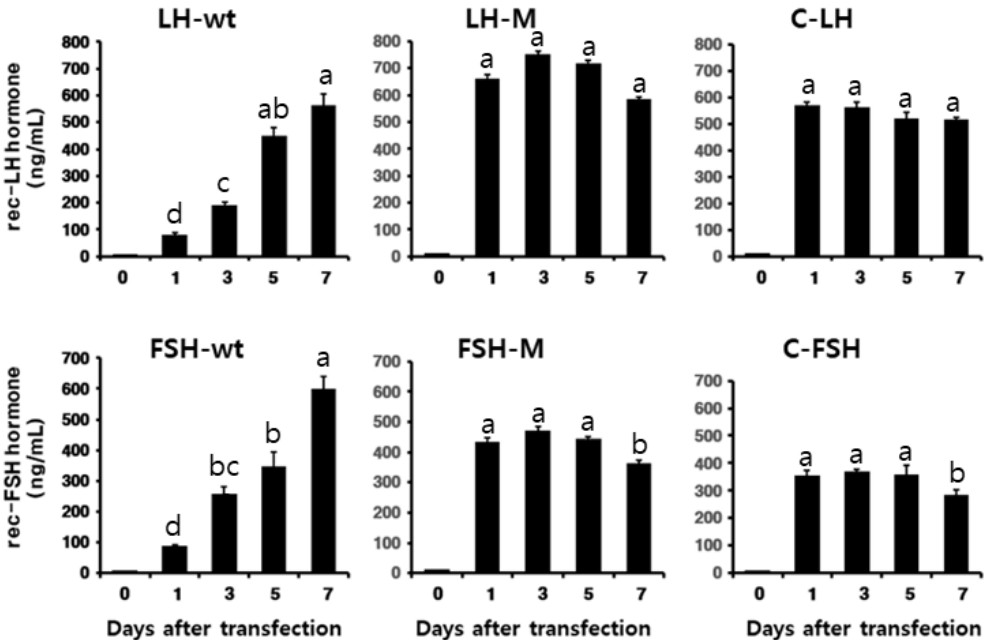

**Figure 2.** Concentrations of rec-eel LHβ/α-wt, eel FSHβ/α-wt, and mutant proteins secreted in the CHO suspension cell culture over time. The expression vectors were transfected into CHO-suspension cells, and the culture medium was collected and centrifuged on days 0, 1, 3, 5, and 7 post-transfections. The expression level of rec-proteins was analyzed using sandwich enzyme-linked immunosorbent assay, as described in the Section 2. Values are expressed as mean $\pm$ standard error of mean from at least three independent experiments. Values with different superscripts were significantly different ($p < 0.05$).

These results indicate that the O-linked glycosylation sites at the eCG β-subunit specifically affected the secretion into the CHO-S cell culture medium, irrespective of the addition in the middle of the β- and α-subunit or in the N-terminal region. Thus, the functions of the eCTP region, including the several O-linked glycosylation sites, need to be studied in a more systematic manner.

*3.2. Deglycosylation of the Rec-Eel LH and FSH Proteins*

Next, the molecular weight of the rec-eel FSH mutants was determined using Western blotting with the anti-eel FSH monoclonal antibody (5A14), previously reported from our laboratory [27]. Western blot analysis revealed an approximate molecular weight of 34 kDa for rec-eel FSHβ/α-wt produced in CHO-S cells (Figure 3a). PNGase F treatment of rec-eel FSHβ/α-wt reduced the molecular weight to approximately 26 kDa, showing a decrease of approximately 7–8 kDa. The Western blotting for rec-eel LHβ/α-wt also revealed that the molecular weight was slightly reduced to approximately 32 kDa. PNGase F treatment of rec-eel LHβ/α-wt yielded similar results as observed for FSHβ/α-wt, with the molecular weight of the protein reducing to approximately 26 kDa.

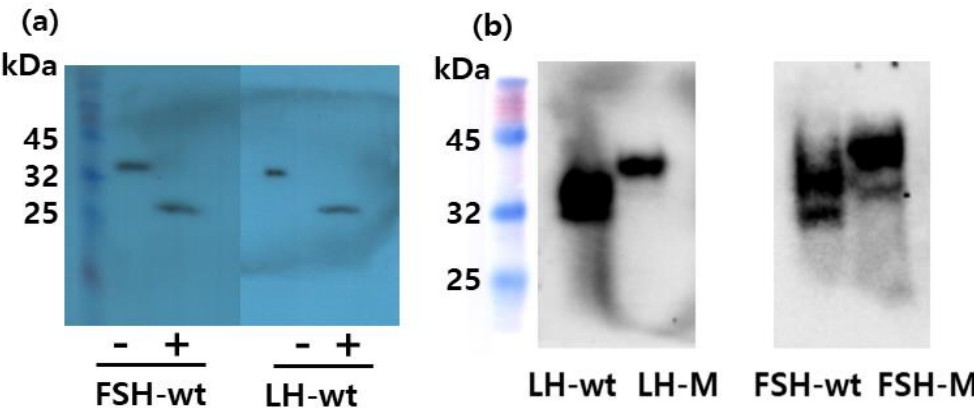

**Figure 3.** Western blot analysis of rec-eel LHβ/α-wt, eel FSHβ/α-wt, LH-M, and FSH-M proteins. The conditioned medium was collected on days 7 post-transfection of CHO-suspension cells and concentrated 5–10 fold. The rec-protein samples were resolved using sodium dodecyl sulfate-polyacrylamide gel electrophoresis and blotted onto a membrane. The proteins were detected with a primary monoclonal antibody (anti-eel FSH5A5 against LH and FSH5A14 against FSH) and a secondary horseradish peroxidase-conjugated goat anti-mouse IgG antibody. The proteins were treated with peptide-N-glycanase F to remove N-linked oligosaccharides and subjected to Western blotting. (**a**) In total, 10 μg of the eel FSHβ/α-wt and eel LHβ/α-wt proteins were loaded in the wells. Lane 1, Marker; Lanes 2 and 3, rec-eel FSHβ/α-wt; Lanes 3 and 4, rec-eel LHβ/α-wt. (**b**) A total of 30 μg of proteins was loaded in the wells. Lane 1, Marker; Lane 2, rec-eel LHβ/α-wt; Lane 3, rec-eel LH-M; Lane 4, FSHβ/α-wt; Lane 5, FSH-M; "−" not treated with peptide-N-glycanase F; "+" treated with peptide-N-glycanase F.

Next, we compared the molecular weights of rec-eel LH-M and FSH-M with those of the corresponding wt proteins. The molecular weights of LH-M (40 kDa) and FSH-M (42 kDa) were higher than those of the respective wild-type proteins (Figure 3b). Two bands in the rec-eel FSH-wt and FSH-M are presumed to be the cause of the difference in the modified oligosaccharide chains. The molecular weight of the mutants was increased approximately by 8 kDa compared with that of both the wild types. For further characterization, we digested rec-eel FSH-M and eel LH-M with PNGase F (Figure 4a). The molecular weights of rec-eel LH-M and eel FSH-M proteins were remarkably decreased to 32 and 34 kDa, which were similar to the molecular weights of eel LHβ/α-wt and eel FSHβ/α-wt, respectively. The molecular weights of C-LH and C-FSH were very similar to those of LH-M and FSH-M (Figure 4b). The eCTP region has only 35 amino acids and includes 11–12 potential O-linked glycosylation sites. We contend that the 8 kDa increase in the molecular weight is due to the additional oligosaccharides at the O-linked glycosylation sites. Thus, we suggest that the eCTP-containing mutants are modified with oligosaccharides in a similar manner, irrespective of the presence of the sites in the N-terminal region or in the middle of the β- and α-subunit.

Next, we compared the rec-eel LH and FSH proteins produced previously in our laboratory in the baculovirus system [27]. The molecular weight was approximately 34–35 kDa. However, the molecular weight was only slightly decreased (by 5–7 kDa) upon the PNGase F treatment (Figure 4c). Thus, our results indicate that the post-translation oligosaccharide modification is less developed in the baculovirus system compared with that in the CHO-S cell system. However, it is difficult to determine the precise molecular mass for the carbohydrate content on SDS gels due to the presence of broad bands because of the attached oligosaccharides.

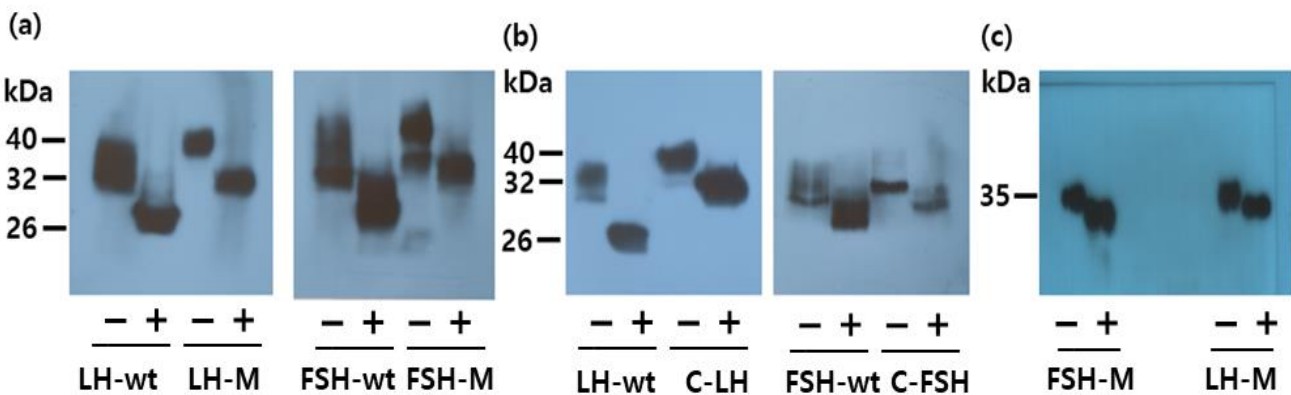

**Figure 4.** Western blot analysis of single-chain forms and mutant proteins, including rec-eel LHβ/α-wt and eel FSHβ/α-wt. Conditioned medium from transfected CHO-suspension cells was prepared for SDS-PAGE. (**a**) LH-M and FSH-M. (**b**) C-LH and C-FSH. (**c**) rec-LH-M and rec-FSH-M proteins produced using the baculovirus system. The samples were electrophoresed and detected with specific monoclonal antibodies (anti-eel FSH5A5 against LH and FSH5A14 against FSH). The proteins were treated with peptide-N-glycanase F to remove N-linked oligosaccharides and subjected to Western blotting. "−" not treated with peptide-N-glycanase F; "+" treated with peptide-N-glycanase F.

### 3.3. Biological Activities of the Variants

The in vitro biological activities of the mutants were assessed using transfected CHO-K1 cells expressing the eel LH and FSH receptors, as reported previously [11]. The potency of cAMP activation by the rec-eel LH and eel FSH mutants is shown in Figure 5. The concentration–response curves for the mutants were shifted to the left compared with that of the wild type (Figure 5).

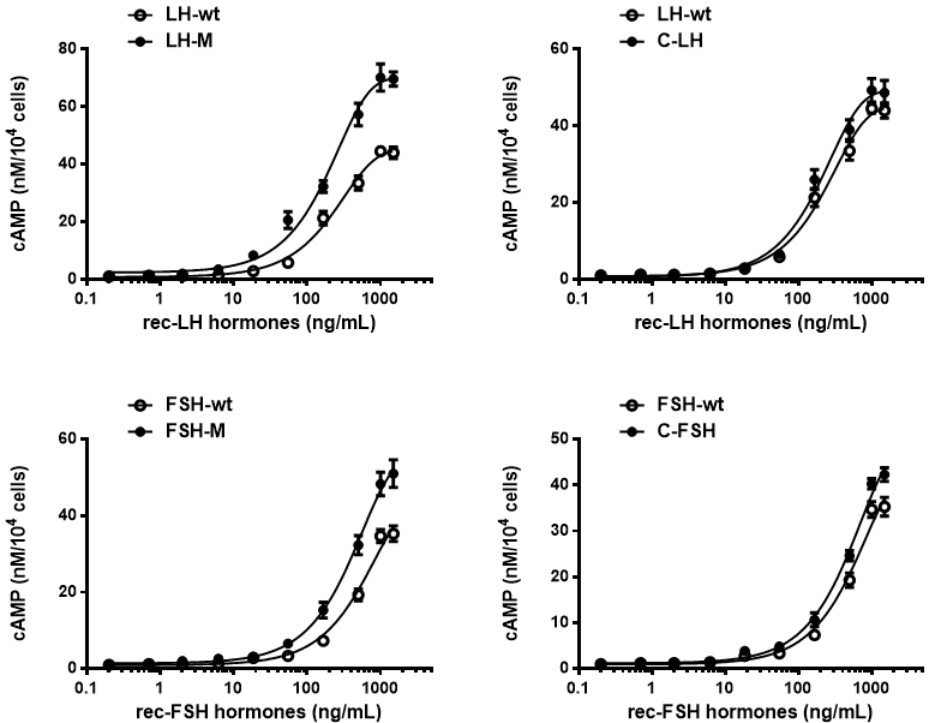

**Figure 5.** Effect of wild-type and mutant recombinant eel LH and eel FSH on the production of cyclic adenine monophosphate (cAMP) in cells expressing the eel lutropin/chorionic gonadotropin receptor (eel LH/CGR) and eel follicle-stimulating hormone receptor (eel FSHR), respectively. CHO-K1 cells transiently transfected with eel LH/CGR or eel FSHR were seeded in 384-well plates (10,000 cells per

well) at 24 h post-transfection. The cells were incubated with re-eel LH or rec-eel FSH for 30 min at room temperature. cAMP production was detected using a homogeneous time-resolved fluorescence assay and is represented as Delta F%. The cAMP concentrations were calculated using the GraphPad Prism 6.0 software (San Diego, CA, USA). The results of the mock-transfected cells were subtracted from each dataset (see Methods). Each data point represents mean $\pm$ standard error of mean from triplicate experiments. The mean data were fitted to the equation of a one-phase exponential decay curve.

The rec-eel LHβ/α-wt exhibited full biological activity, with an $EC_{50}$ value of 222.2 ng/mL and Rmax of $44.9 \pm 4.6$ nM/$10^4$ cells. The mutant with eCTP inserted in the middle of the β- and α-subunit showed higher activity, with the $EC_{50}$ value and Rmax level being 120% and 1.56-fold, respectively. For the C-LH mutant, the concentration–response curve was slightly shifted to the left; however, the $EC_{50}$ value was 80.9% (Table 1).

**Table 1.** Bioactivity of rec-eel LH proteins in cells expressing the eel LH receptor.

| rec-LH Hormones | cAMP Response | | |
| --- | --- | --- | --- |
| | **Basal** | **Log ($EC_{50}$)** | **Rmax** |
| | **(nmoles/$10^4$ Cells)** | **(ng/mL)** | **(nmoles/$10^4$ Cells)** |
| LH-wt | $0.8 \pm 0.4$ | 222.2 <br> (194.5–259.1) | $44.9 \pm 0.9$ |
| LH-M | $1.2 \pm 0.7$ | 184.7 <br> (160.5–217.5) | $70.1 \pm 0.7$ |
| C-LH | $0.6 \pm 0.5$ | 274.5 <br> (163.7–227.1) | $49.2 \pm 1.1$ |

Values are means $\pm$ SEM of data from triplicate experiments. Log ($EC_{50}$) values were determined from the concentration–response curves from in vitro bioassays. Average basal cAMP level without agonist treatment. Rmax average cAMP level/$10^4$ cells. Geometric mean (95% confidence limit) of values from at least three experiments.

The rec-eel FSHβ/α-wt also exhibited full biological activity, with an $EC_{50}$ value of 566.7 ng/mL and Rmax of $44.0 \pm 0.4$ nM/$10^4$ cells. For eel FSH-M, the biological activity was higher than that for FSHβ/α-wt, with $EC_{50}$ value and Rmax of 390.2 ng/mL and $55.7 \pm 1.6$ nM/$10^4$ cells, respectively. The $EC_{50}$ and Rmax values were 145.2% and 1.26-fold, showing that the FSH-M mutant with the eCTP glycosylation sites exhibited higher biological activity. The C-FSH mutant also showed higher biological activity in the cAMP response (117.9%) and Rmax level (49.2 nM/$10^6$ cells) than that observed for FSHβ/α-wt (Table 2).

The $EC_{50}$ value and Rmax level indicated a remarkably increased activity upon addition of the eCTP region in the middle. However, the C-LH and C-FSH mutants with the attachment in the N-terminal region resulted in a slightly increased or decreased activity, and the Rmax levels were very similar to that of the wild type. This is clearly reflected by the significant difference in the bioactivity of LH-M, C-LH, FSH-M, and C-FSH. The LH-M and FSH-M mutants had 120–145% higher bioactivity than that of the wild type. The maximum increase in the bioactivity of the LH-M and FSH-M mutants was approximately 126–156% of that observed for the wild type. Therefore, the rec-eel LH-M and eel FSH-M mutants are critical for signal transduction through their receptors. As a result, the eCTP insertion in the middle of the glycoprotein hormones could play a pivotal role in signal transduction, highlighting the need to produce large amounts of rec-proteins in a stable expression system.

**Table 2.** Bioactivity of rec-eel FSH proteins in cells expressing the eel FSH receptor.

| rec-FSH Hormones | cAMP Response | | |
|---|---|---|---|
| | Basal | Log (EC$_{50}$) | Rmax |
| | (nmoles/10$^4$ Cells) | (ng/mL) | (nmoles/10$^4$ Cells) |
| FSH-wt | 0.7 ± 0.2 | 730.2 (512.6–1269) | 52.9 ± 1.8 |
| FSH-M | 1.2 ± 0.3 | 425.1 (371.1–497.5) | 59.6 ± 1.7 |
| C-FSH | 0.9 ± 0.5 | 567.6 (403.6–956.2) | 49.9 ± 0.6 |

Values are means ± SEM of data from triplicate experiments. Log (EC$_{50}$) values were determined from the concentration–response curves from in vitro bioassays. Average basal cAMP level without agonist treatment. Rmax average cAMP level/10$^4$ cells. Geometric mean (95% confidence limit) of values from at least three experiments.

### 3.4. Characterization of the Other Mutants with eCTP and Myc-tag

We constructed other mutants with insertion in the C-terminal region of single chain β/α (designed as LH-C and FSH-C). LH-Myc-C and FSH-Myc-C, with Myc-tag inserted between the β- and α-subunit, were constructed, and their biological activity and expression level were analyzed (Figure 6).

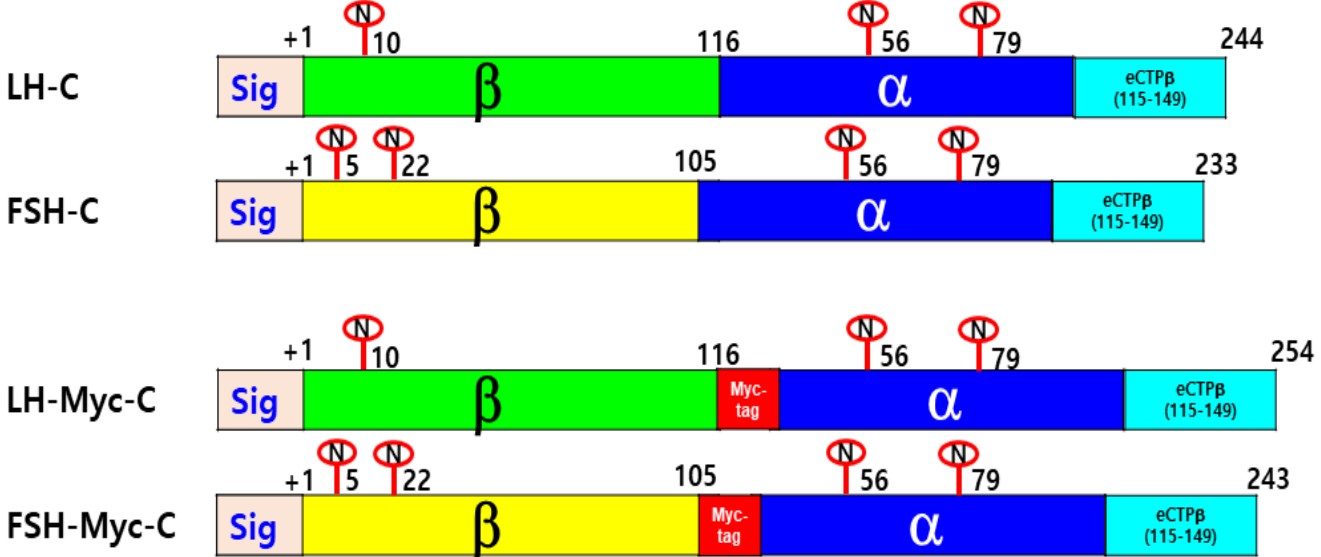

**Figure 6.** Schematic diagram of rec-eel LH-C, FSH-C, LH-Myc-C, and FSH-Myc-C mutants. The eCTP region was inserted into the C-terminal regions of eel LHβ/α-wt and FSHβ/α-wt. The Myc-tag (10 amino acids) was also attached between the β- and α-subunit. The mutants were designated as pcDNA3-eel LH-C, FSH-C, LH-Myc-C, and FSH-Myc-C. The numbers indicate the amino acids in the mature proteins. The encircled "N" denotes N-linked oligosaccharide on the eel LH, eel FSH, and mutants, respectively. The numbers in addition to the encircled "N" indicate the N-linked glycosylation sites. The eCTPβ (115–149) region contains potential O-linked oligosaccharide sites. The *myc*-tag was added between the β-subunit and α-subunit in the LH-C and FSH-C mutants. The green color is LHβ, the yellow color is FSHβ, α-subunit is blue color, and the skyblue color is the eCTPβ.

The LH-C mutant was not detected in Western blotting with the monoclonal antibody (Figure 7a). However, the FSH-C mutant protein was detected, as were FSH-M and C-FSH (Figure 4a). In the Myc-tag attachment, the LH-Myc-C mutant was weakly detected by the Myc-tag monoclonal antibody. However, the FSH-Myc-C mutant did not give any

signal with the same antibody, but a distinct band was detected by the 5A14 antibody (Figure 7b). Thus, we could not measure the secreted quantity of the LH-C and FSH-C using the monoclonal antibody developed in our laboratory.

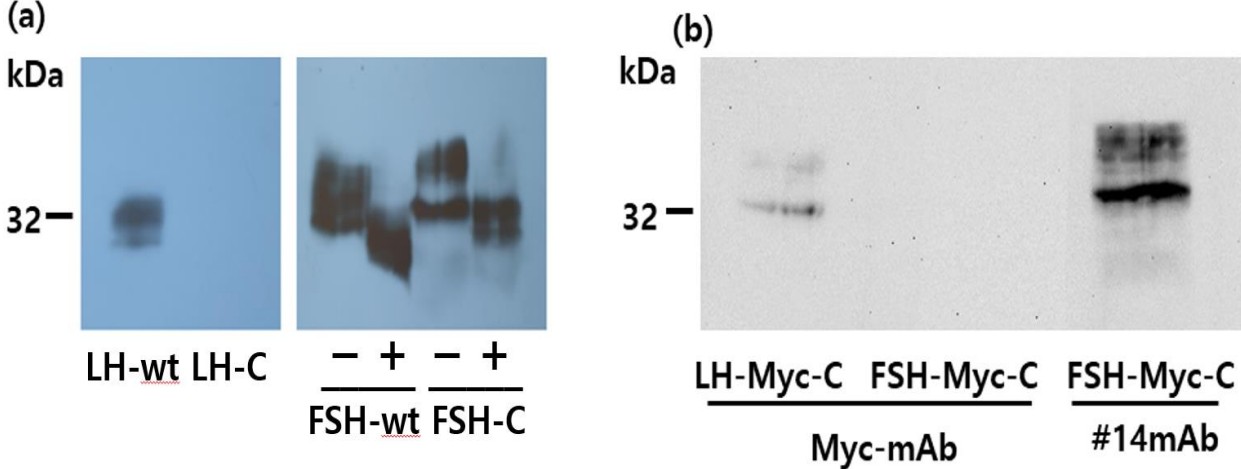

**Figure 7.** Western blot analysis of rec-eel LH-C, FSH-C, LH-Myc-C, and FSH-Myc-C mutants. Conditioned medium from transfected CHO-suspension cells was prepared for SDS-PAGE, as described in Materials and Methods. (**a**) LH-C and FSH-C. (**b**) LH-Myc-C and FSH-Myc-C. The rec-proteins in panel a were detected with specific monoclonal antibodies (anti-eel FSH5A5 against LH and FSH5A14 against FSH). The proteins were treated with peptide-N-glycanase F to remove N-linked oligosaccharides and subjected to Western blotting. The rec-proteins in pannel b were detected with an anti-Myc monoclonal antibody (lanes 1 and 2). For the band in lane 3, detection was performed with a specific monoclonal antibody against eel FSHβ/α (FSH5A14). "−" not treated with peptide-N-glycanase F; "+" treated with peptide-N-glycanase F.

## 4. Discussion

The present study indicates that eel LH and eel FSH mutants exhibit more potent activity through their receptors. The variants of single chain showed increased activities upon insertion of 35 amino acids of the eCTP region, indicating that the inserted linker sequence resulted in the most suitable structure of the protein for biological activity. In glycoprotein hormones, the eCTP region would be essential for early expression and potent biological activity of single chains.

The β-subunit of eCG contains one N-linked glycosylation site as Asn[13] and approximately 11–12 O-linked glycosylation sites in the CTP region [3,28,29]. Among the known glycoprotein hormones, eCG has the highest (more than 40%) carbohydrate content [30,31]. Therefore, we chose the eCTP region to design eel LH and eel FSH analogs displaying potent activity. Recently, we reported that the glycosylation sites play an essential role in the biological activities of eel LH [3], eel FSH [21], and eCG [2,3]. In particular, the specific glycosylation sites in eel LHβ/α-wt and eel FSHβ/α-wt play a pivotal role in the biological activity through their receptors as well as in their secretion into the supernatant.

In the present study, we characterized the secretion and signal transduction of the eel LH and eel FSH mutants. The expression of all mutants with the 35 amino acids of the eCTP region, containing the 11–12 potential O-linked glycosylation sites, peaked on day 1 post-transfection; in contrast, the expression of the wild-type proteins was very low. Our data are consistent with those in our previous study, indicating that the deletion mutant (eCGβ-D/α, eCGβ-D/αΔ56) of the CTP in eCGβ/α has greatly delayed secretion [2]. In studies on hLH, the hLH β-subunit attached to the hCG β-CTP region [32] and hTSH [25,33] were detected early in the culture medium. The deletion of the hCTP region in hCGβ/α delayed the secretion into the medium [26]. Thus, we suggest that the CTP regions in the C-terminal region of the eCG β-subunit are indispensable for early secretion in mammalian cells.

The eel α-subunit purified from the pituitary migrates at 17 and 19 kDa, but the molecular weight was decreased to 13 kDa upon N-glycopeptidase F treatment. The FSH β-subunit was detected as two bands between 21 and 17 kDa [34]. Thus, the molecular weight of eel LH and FSH from the eel pituitary was determined to be approximately 30–40 kDa [34]. The purified rec-eel LH α-subunit and β-subunit were detected as single peptides with molecular weights of 17.5 and 16.5 kDa, respectively, in the Drosophila S2 cells [23]. In the methylotrophic yeast *Pichia pastoris*, the molecular weight of rec-eel FSH β-subunit was 16.4 and 26.4 kDa [35]. Therefore, the molecular weight of eel LH and FSH glycoprotein hormones could be slightly changed depending on the host cell.

In the present study, the molecular weight of rec-eel LHβ/α-wt and eel FSHβ/α-wt was approximately 32 and 34 kDa, and PNGase treatment decreased it to 26 kDa. The molecular weights of eel LH-M, C-LH, FSH-M, and C-FSH mutants were 40 and 42 kDa, respectively. In previous studies, modified oligosaccharides were detected as broad bands in mammalian cells [2,3]. In the production of rec-eel gonadotropin using the baculovirus system, a broad band was detected at 30–35 kDa for rec-eel LH and rec-eel FSH with the insertion of a 34-amino acid eCTP linker [24]. Although the eel LH β-subunit has 10 extra amino acids than the eel FSH β-subunit, the reason for the slighter reduction in the molecular weight of rec-LHβ/α-wt was assumed to be the difference in the number of glycosylation sites. Only one glycosylation site at the $Asn^{10}$ is present in the LH β-subunit, unlike two sites at the $Asn^5$ and $Asn^{22}$ in the FSH β-subunit.

Thus, we surmise that expression in CHO-S cells is associated with more glycosylation sites than that in the baculovirus system. These results suggest that the rec-eel LH and FSH proteins have different oligosaccharide modifications depending on the cell type.

In a previous study using rec-human LHβ/α (hLH), the mutant with an insertion of 31 amino acids of the hCG β-subunit CTP was detected to have a molecular weight of approximately 5–6 kDa [32]. The hTSHβ+hCGβCTP region+α, added hCGβCTP region as a linker in between the hTSH β- and α-subunit, was detected as a heavier band because of the additional protein sequence and the attachment of O-linked oligosaccharides on the CTP [36]. Our results are consistent with those of previous studies, suggesting that the molecular weights were dramatically increased upon attachment of the eCTP linker. We detected the bands of eel LH-M, C-LH, FSH-M, and C-FSH mutants at 40 and 42 kDa, respectively. However, PNGase treatment of mutants clearly decreased the molecular weights to 32 and 34 kDa. The molecular weight of the mutants was increased by approximately 8 kDa, demonstrating that the attachment of the linker, including O-linked glycosylation sites, resulted in the increase in molecular weight.

We and others have shown that ligands (hCG, eCG, eFSH, eel FSH, and eel LH) of glycoprotein hormone receptors result in cAMP responses through their receptors [1,2,37]. Many researchers have reported a 5–10-fold reduction in biological activity upon removal of the glycosylation sites [2,3,19]. In particular, deglycosylation of the glycoprotein hormones at the specific sites reduced cAMP responsiveness at the $Asn^{52}$ site of the hCG α-subunit [19], hTSH [17], hFSH [16,18,20], and at the $Asn^{56}$ site of the eCG α-subunit [3]. In the eel glycoprotein hormones, we reported that the glycosylation site at $Asn^{56}$ in the eel LH α-subunit and at $Asn^{10}$ in the LH β-subunit did not show any signal transduction ability [1]. The deglycosylation mutants of eel FSH exhibited a dramatic decrease in the biological activity with regard to the $EC_{50}$ value and maximum cAMP stimulation [1]. Thus, we suggest that the specific N-linked glycosylation sites are indispensable for signal transduction through their receptor–ligand complexes. We also show that the mutants containing eCTP in the N-terminal region and in the middle region display higher biological activity than that of the wild type. Our data are consistent with those of previous studies on hFSH [38,39], indicating that the oligosaccharide sites play an important role in determining the biological activity of eel glycoprotein hormones. These studies have allowed the identification of the role of glycosylation in eel LH and FSH.

We believe that the insertion of the eCTP region in the middle of eel LHβ/α-wt and FSHβ/α-wt plays a pivotal role for signal transduction through their respective receptors

and for secretion into the CHO-S cell culture supernatant. We indicate that the eCTP insertion in the middle site of LHβ/α and FSHβ/α could lead to massive production of rec-glycoproteins hormones in the stable expression system of CHO-DG44 and Freedom™ CHO-S cells. We suggest that the eCTP specific linker for rec-eel LH and eel FSH enhances the overall stability in vitro, and the linker attachment is required for maximal cAMP responsiveness, which indicates that it is critical for efficient secretion and is likely to be important for biological activity. In the present study, although we did not compare in vivo experiment both eel LH-wt, eel FSH-wt and mutants, these novel rec-eel LH and FSH proteins exhibiting potent activity reported in this study could provide potentially valuable models to induce eel maturation in vivo.

## 5. Conclusions

Eel LH and FSH mutants containing the eCG β-subunit CTP region were efficiently secreted, and their expression time was faster than that of LH-wt and FSH-wt, with the expression peaking at 1-day post-transfection and being consistently maintained until the final collection. The molecular weights of the LH-M and FSH-M mutants were 40–43 kDa and 42–45 kDa, respectively. The molecular weight of deglycosylated proteins was considerably decreased by approximately 8–10 kDa. The $EC_{50}$ values and the maximal responsiveness of the mutants were increased by approximately 1.2- to 1.4-fold and 156% and 126% of that observed for both the LH-wt and FSH-wt proteins. These results provide evidence that the C-terminal region of the eCG β-subunit plays a pivotal role in early secretion and signal transduction in a mammalian expression system. We suggest that these novel rec-eel LH and FSH proteins expressing the exhibiting potent activity could be produced in large quantities using the stable CHO cell system.

**Author Contributions:** Conceptualization, K.-S.M.; methodology, M.B.; formal analysis, S.-G.K.; data curation, M.B. and K.-S.M.; software, S.H.P.; investigation, M.B.; writing—original draft preparation, M.B.; writing—review and editing, M.-H.K. and K.-S.M.; project administration, M.B., funding acquisition, M.G.S. and S.-K.K. All authors have read and agreed to the published version of the manuscript.

**Funding:** This work was supported by a research grant from the National Institute of Fisheries Science (R2023026), Republic of Korea.

**Institutional Review Board Statement:** Not applicable.

**Informed Consent Statement:** Not applicable.

**Data Availability Statement:** Data are contained within the article.

**Acknowledgments:** The authors thank HW Seong for helpful discussion.

**Conflicts of Interest:** The authors declare no potential conflict of interest.

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
