# Peer review of "Production of Recombinant Single-Chain Eel Luteinizing Hormone and Follicle-Stimulating Hormone Analogs in Chinese Hamster Ovary Suspension Cell Culture"

_cimb, doi:10.3390/cimb46010035_

Round 1
Reviewer 1 Report
Comments and Suggestions for Authors
The current manuscript presents novel wild-type and various mutant forms of recombinant eel gonadotropin hormones (LH and FSH) expressed in Chinese hamster ovary suspension cells. Mutant forms involved the insertion of O-linked glycosylated carboxyl-terminal peptide from the equine chorionic gonadotropin β-subunit (eCTP) between the β- and α-subunits or in the N-terminal or C-terminal regions. The study establishes that eCTP plays a crucial role in the early secretion and signal transduction of gonadotropins within a mammalian expression system, demonstrating its potent biological activity.
1. At line 180, “10X” is incorrect, the symbol "×" should be replaced with the letter X.
2. In Figure 2, the vertical axes of all six plots should be clearly labeled to indicate rec-LH, rec-FSH, LH-M, FSH-M, C-LH, and C-FSH.
3. Lines 251-255 and 373-376 are both parts of the discussion and they are not recommended to be placed in the Results section, but in the Discussion section.
4. In the figure legends of Figure 3 (lines 265 and 266), there are two symbol errors, they are supposed to be semicolons.
5. In line 467, "in vitro" should be italicized.
6. The manuscript emphasized the heightened biological activity of rec-eel FSH/LH post eCTP insertion compared to the non-mutant type. However, it lacked an experimental comparison of rec-eel and mammalian gonadotropin activity and function. As a result, the broader applications of these novel rec-eel FSH/LH in mammalian systems have not been definitively established.
7. The manusctipt concludes that "The eel LH-M and FSH-M mutants could provide potentially valuable models to induce eel maturation in vivo and help elucidate the structure–function relationship." However, all experiments in this study were validated in mammalian cell lines, and it cannot be assumed that they have the same effect in non-mammalian species like eels. Functional verification in relevant cell lines or animals is necessary before drawing such conclusions.
Author Response
Reviewer 1
The current manuscript presents novel wild-type and various mutant forms of recombinant eel gonadotropin hormones (LH and FSH) expressed in Chinese hamster ovary suspension cells. Mutant forms involved the insertion of O-linked glycosylated carboxyl-terminal peptide from the equine chorionic gonadotropin β-subunit (eCTP) between the β- and α-subunits or in the N-terminal or C-terminal regions. The study establishes that eCTP plays a crucial role in the early secretion and signal transduction of gonadotropins within a mammalian expression system, demonstrating its potent biological activity.
- At line 180, “10X” is incorrect, the symbol "×" should be replaced with the letter X.
→10X changed to the 10X at the Line 184-185.
- In Figure 2, the vertical axes of all six plots should be clearly labeled to indicate rec-LH, rec-FSH, LH-M, FSH-M, C-LH, and C-FSH.
→In the Figure 2, we inserted the each mutant name on the upper.
- Lines 251-255 and 373-376 are both parts of the discussion and they are not recommended to be placed in the Results section, but in the Discussion section.
→We moved both sentences to discussion section by reviewer’s comment.
- In the figure legends of Figure 3 (lines 265 and 266), there are two symbol errors, they are supposed to be semicolons.
→We changed by the reviewer’s comment.
- In line 467, "in vitro" should be italicized.
→We changed “in vitro’ to “in vitro”.
- The manuscript emphasized the heightened biological activity of rec-eel FSH/LH post eCTP insertion compared to the non-mutant type. However, it lacked an experimental comparison of rec-eel and mammalian gonadotropin activity and function. As a result, the broader applications of these novel rec-eel FSH/LH in mammalian systems have not been definitively established.
→We inserted “In the present study, although we did not compare in vivo experiment both eel LH-wt, eel FSH-wt and mutants, the rec-LH and FSH mutant proteins produced from CHO-DG 44 cells are showing a certain degree of the biological activity” in the final discussion section by the reviewer’s comments.
- The manuscript concludes that "The eel LH-M and FSH-M mutants could provide potentially valuable models to induce eel maturation in vivoand help elucidate the structure–function relationship." However, all experiments in this study were validated in mammalian cell lines, and it cannot be assumed that they have the same effect in non-mammalian species like eels. Functional verification in relevant cell lines or animals is necessary before drawing such conclusions.
→We deleted “The eel LH-M and FSH-M mutants could provide potentially valuable models to induce eel maturation in vivo and help elucidate the structure–function relationship“ by the reviewer’s comment.

Reviewer 2 Report
Comments and Suggestions for Authors
The present paper by M. Byambaragchaa et al deals with the production in CHO cells, and the in vitro characterization of various single-chain derivatives of eel FSH and eel LH.
General comment :
Although this report addresses an interesting issue, it is too superficial and contains too many approximations to be convincing. In particular, the naming of the different hormones is misleading. « LH-wt » or « FSH-wt » suggest that wild-type subunits were co-transfected to get heterodimeric eelLH and eel FSH. So sc-LH-wt or sc-FSH-wt would be better although not strictly sufficient (alpha-beta or beta-alpha). Many papers dealing with sc-eCGs or other single-chain gonadotropins are not mentioned.
Specific comments :
The title contains several assertions or oblivions that are misleading. 1/ It does not mention that ALL hormones under study are single-chain molecules 2/The term « Superactive » is exaggerated and imprecise as only in vitro estimates of bioactivity were done and no well-identified reference preparation is mentioned in the text. Moreover, the hormones should be named either « Lutropin and Follitropin » or « Luteinizing Hormone and Follicle-Stimulating Hormone » (not a mix of both kinds of names as in the present title).
lines 53-54 : it should be clearly mentioned that the previous studies only addressed the characterization of recombinant heterodimeric eel LH and FSH molecules and not natural gonadotropins from eel pituitaries.
Figure 1 : The N-saccharide positions in the mutant sc-LH and sc-FSH should be shown as well. They are not absent. The O-glycosylations of the eCTPβ sequence should also be shown as they introduce a large shift in MW.
Maybe the sc-LH-C and sc-FSH-C schemes should be shown in this figure together with those of sc-C-LH sc-C-FSH for easier comparison.
lines 146-155 : This chapter is rather confusing.
a/ What are the reference preparations used to quantify the amounts of the various sc-eelFSH and sc-eelLH produced ? Are they the heterodimeric wt-eelLH and wt-eelFSH ?
b/ It is weird (l. 149) to describe the FSH5A5 antibody as an anti-LHα. Anti α or eelGPHα would be more adequate.
c/ It seems that the two monoclonal antibodies used to quantify the sc-eelLHs are both antibodies against eelFSHα and thus also against eelLHα. This quantitation does not ascertain the presence of a correctly-folded eelLHβ domain.
line 215 : It should be mentioned that the quantitations of the secretion of recombinant sc-hormones are performed by ELISA.
Figuer 2 : This figure and its caption are incomplete.
a/ The six panels should be clearly identified
b/ The concentrations of secreted hormones are expressed in ng/mL but the reference preparations are not indicated
c/ Typical dose-response curves for the various sc-hormones and the reference preparation should be presented to ascertain the quality of the determinations.
d/ It is not clear whether the recorded secreted hormone concentrations each day are from day 0 (d0 > d1 ; d0 > d3 ; d0 > d5 ; d0 > d7) or from the previous determination (d0 > d1 ; d1 > d3 ; d3 > d5 ; d5 > d7).
lines 222-227 : The meaning of the presented values is difficult to evaluate as the reference preparation used is not indicated. This reference should preferentially be natural pituitary eel LH and eel FSH or, if not available, heterodimeric wild-type eel LH and FSH.
line 252 : « slighter » instead of « slight »
Figure 3 :
a/ It should be indicated in the caption that the loaded quantities (10 and 30µg) correspond to the values determined by ELISA.
b/ The sc-LH-M band is much sharper than that of sc-LH-wt. It could be that the real quantity in weight of the former is much higher than that of the latter (maybe because the latter is more active in ELISA than the former).
c/ Is there an explanation for the presence of two bands for both sc-FSH-wt and sc-FSH-M ?
Figure 5 :
a/ It should be indicated that the quantities in the abscissa derive from ELISA determinations (and unknown reference).
b/ It would have been interesting to test the sc-LHs in the FSH bioassay and sc-FSHs in the LH bioassay to test the specificity of the assays.
c/ If possible it would be interesting to fuse tables 1 and 2 with Figure 5
Tables 1 and 2 The significance of values indicated as « nM/104 cells » without the volume indicated. Do you mean « nmoles/104 cells » ?
Figure 6 caption : The eCTP sequence cannot be called « linker » when at the C-terminus of the sc-molecules.
Line 369 : C-FSH is not shown in Figure 7A.
Lines 373-376 and 387-389 should be moved to the Discussion section
Line 433 « more glycosylation » is unclear ; « more glycosylation sites » or « longer glycosyl chains ».
The conclusion is hypothetical and does not directly derive from the shown data.
Author Response
Reviewer 2
General comment:
Although this report addresses an interesting issue, it is too superficial and contains too many approximations to be convincing. In particular, the naming of the different hormones is misleading. « LH-wt » or « FSH-wt » suggest that wild-type subunits were co-transfected to get heterodimeric eelLH and eel FSH. So sc-LH-wt or sc-FSH-wt would be better although not strictly sufficient (alpha-beta or beta-alpha). Many papers dealing with sc-eCGs or other single-chain gonadotropins are not mentioned.
Specific comments:
The title contains several assertions or oblivions that are misleading. 1/ It does not mention that ALL hormones under study are single-chain molecules 2/The term « Superactive » is exaggerated and imprecise as only in vitro estimates of bioactivity were done and no well-identified reference preparation is mentioned in the text. Moreover, the hormones should be named either « Lutropin and Follitropin » or « Luteinizing Hormone and Follicle-Stimulating Hormone » (not a mix of both kinds of names as in the present title).
→We deleted “superactive” in the title and changed “lutropin” to luteinizing hormone.
lines 53-54: it should be clearly mentioned that the previous studies only addressed the characterization of recombinant heterodimeric eel LH and FSH molecules and not natural gonadotropins from eel pituitaries.
→We inserted “recombinant (rec)” in the same Line.
Figure 1: The N-saccharide positions in the mutant sc-LH and sc-FSH should be shown as well. They are not absent. The O-glycosylations of the eCTPβ sequence should also be shown as they introduce a large shift in MW.
Maybe the sc-LH-C and sc-FSH-C schemes should be shown in this figure together with those of sc-C-LH sc-C-FSH for easier comparison.
→We inserted “N-linked oligosaccharide sites in the LH and FSH”. And we also inserted “eCGb-subunit amino acids”.
lines 146-155: This chapter is rather confusing.
a/ What are the reference preparations used to quantify the amounts of the various sc-eelFSH and sc-eelLH produced ? Are they the heterodimeric wt-eelLH and wt-eelFSH ?
b/ It is weird (l. 149) to describe the FSH5A5 antibody as an anti-LHα. Anti α or eelGPHα would be more adequate.
c/ It seems that the two monoclonal antibodies used to quantify the sc-eelLHs are both antibodies against eelFSHα and thus also against eelLHα. This quantitation does not ascertain the presence of a correctly-folded eelLHβ domain.
→a/We mentioned the eel LH-wt and eel FSH-wt as shown in the Figure 1. It is single chain rec-protein.
→b/We reported monoclonal antibodies produced in the previous paper by the reference 32, 33. FSH5A5 binds to the a-subunit of eel LH and FSH5A14 binds to the FSH b-subunit of eel.
→c/We explained the anti-eel FSH 5A11 labelled with HRP. Thus, we used 3 specific monoclonal antibodies for ELISA assay as our previous report.
line 215: It should be mentioned that the quantitations of the secretion of recombinant sc-hormones are performed by ELISA.
→We inserted “The quantitation of the secretion of rec-hormones was performed by ELISA)” in the same Line.
Figuer 2 : This figure and its caption are incomplete.
a/ The six panels should be clearly identified
b/ The concentrations of secreted hormones are expressed in ng/mL but the reference preparations are not indicated
c/ Typical dose-response curves for the various sc-hormones and the reference preparation should be presented to ascertain the quality of the determinations.
d/ It is not clear whether the recorded secreted hormone concentrations each day are from day 0 (d0 > d1 ; d0 > d3 ; d0 > d5 ; d0 > d7) or from the previous determination (d0 > d1 ; d1 > d3 ; d3 > d5 ; d5 > d7).
→a/We inserted “LH-wt, LH-M, C-LH, FSH-wt, FSH-M and C-FSH” on the upper.
→b,c/We suggested “standard sample (0-400 ng/mL) purified from rec-eel FSH in Ecoli” in the 2.4 section of Materials and Methods.
→d/We determined that d0 means the transfected day and d1 indicates the next day after transfection
lines 222-227: The meaning of the presented values is difficult to evaluate as the reference preparation used is not indicated. This reference should preferentially be natural pituitary eel LH and eel FSH or, if not available, heterodimeric wild-type eel LH and FSH.
→We suggested that rec-eel LHs and -eel FSHs measured by ELISA. In the our previous paper, we used to standard sample as the rec-eel FSH b/a expressed in E.coli and this protein was used to antigen for monoclonal antibody production.
line 252: « slighter » instead of « slight »
→We changed “slight” to “slighter”.
Figure 3:
a/ It should be indicated in the caption that the loaded quantities (10 and 30µg) correspond to the values determined by ELISA.
b/ The sc-LH-M band is much sharper than that of sc-LH-wt. It could be that the real quantity in weight of the former is much higher than that of the latter (maybe because the latter is more active in ELISA than the former).
c/ Is there an explanation for the presence of two bands for both sc-FSH-wt and sc-FSH-M ?
→a/we explained the loading quantity under captions (a) 10 ug and (b) 30 ug.
→c/we inserted “Two bands in the rec-eel FSH-wt and FSH-M are presumed to be the cause of the difference in the modified oligosaccharide chains.”
Figure 5:
a/ It should be indicated that the quantities in the abscissa derive from ELISA determinations (and unknown reference).
b/ It would have been interesting to test the sc-LHs in the FSH bioassay and sc-FSHs in the LH bioassay to test the specificity of the assays.
c/ If possible, it would be interesting to fuse tables 1 and 2 with Figure 5
→a/We developed ELISA system of rec-eel LH and eel FSH analysis in our previous report. Thus, we insisted that our ELISA system is correct by our previous many papers (reference 32 and 33).
→b/We tested the bioactivities in cells expressing those LH/CGR and FSHR. There is no response in the cross-analysis.
→c/We considered the fuse tables 1 and 2 with Figure 5. However we determined the present condition.
Tables 1 and 2 The significance of values indicated as « nM/104 cells » without the volume indicated. Do you mean « nmoles/104 cells » ?
→We changed “nM/104 cells’ to “nmoles”.
Figure 6 caption: The eCTP sequence cannot be called « linker » when at the C-terminus of the sc-molecules.
→We changed “linker” to “region”.
Line 369 : C-FSH is not shown in Figure 7A.
→We changed “Figure 7A” to “Figure 4A”.
Lines 373-376 and 387-389 should be moved to the Discussion section.
→We moved “two sentence” to “discussion section”.
Line 433 « more glycosylation » is unclear; « more glycosylation sites » or « longer glycosyl chains ».
→more glycosylation changed to more glycosylation sites.
The conclusion is hypothetical and does not directly derive from the shown data.
→d/we changed some content in the conclusion.

Reviewer 3 Report
Comments and Suggestions for Authors
In the paper entitled ‘Production of Superactive Recombinant Eel Lutropin and Folicle-Stimulating Hormone Analogs in CHO Suspension Cell 3 Culture’, Authors describes the method of biosynthesis recombinant eel luteinizing hormone (rec-eel LH) and follicle-stimulating hormone (rec-eel FSH), which in turn display high biological activity in Chinese hamster ovary suspension (CHO-S) cells. In general, their work is interesting and quite well-designed. Although these studies are basic, the search for new LH and FSH analogs and the methods of their production may also be important for medicine, because there is an ongoing search for more effective gonadotropin analogs that would be more tolerable for the patient. The topic of the article fits well with the scope of Current Issues in Molecular Biology.
The paper is generally well written, however, I have some comments regarding the presentation of the results obtained and conclusions drawn on their basis:
1 1) Figure 2 – is not legible, I think it should be divided into panels because the way of presenting the results does not allow them to determine which graphs refer to wt gonadotropins and which to mutants. Moreover, based on ‘eye observations’ (without performing statistical analysis), the conclusion that in the CHO suspension cell culture, the expression/ secretion of eel LH and FSH mutants containing eCTP time was faster than that of wild-type is not justified.
22) Figure 5 - refers to a very important part of the results regarding the biological activity of the subjects' wild-type and mutant recombinant eel LH and eel FSH. These results are discussed in the later part of the manuscript, and according to the author's suggestion support their findings that the EC50 and the maximal responsiveness of the mutants were increased observed for both the wild-type gonadotropins. However, in my opinion, to formulate such a statement, it is necessary to conduct a statistical analysis that will confirm the observations made by the authors. Simply comparing of average values of cAMP levels and Rmax average cAMP levels does not allow them to make such conclusions.
To summarize, due to the interesting topic and the authors' effort put into the experimental part of the work, I believe that the manuscript can be accepted for publication after a major revision.
Author Response
In the paper entitled ‘Production of Superactive Recombinant Eel Lutropin and Folicle-Stimulating Hormone Analogs in CHO Suspension Cell 3 Culture’, Authors describes the method of biosynthesis recombinant eel luteinizing hormone (rec-eel LH) and follicle-stimulating hormone (rec-eel FSH), which in turn display high biological activity in Chinese hamster ovary suspension (CHO-S) cells. In general, their work is interesting and quite well-designed. Although these studies are basic, the search for new LH and FSH analogs and the methods of their production may also be important for medicine, because there is an ongoing search for more effective gonadotropin analogs that would be more tolerable for the patient. The topic of the article fits well with the scope of Current Issues in Molecular Biology.
The paper is generally well written, however, I have some comments regarding the presentation of the results obtained and conclusions drawn on their basis:
- 1) Figure 2 – is not legible, I think it should be divided into panels because the way of presenting the results does not allow them to determine which graphs refer to wt gonadotropins and which to mutants. Moreover, based on ‘eye observations’ (without performing statistical analysis), the conclusion that in the CHO suspension cell culture, the expression/ secretion of eel LH and FSH mutants containing eCTP time was faster than that of wild-type is not justified.
→We did the statistical analysis. GraphPad Prism 6.0 (San Diego, Ca, USA) was used to evaluate the differences between samples using one-way analysis of variance, followed by Turkey’s multiple comparison tests. A p-value of < 0.05 was taken to indicate a significant between the groups.
2 Figure 5 - refers to a very important part of the results regarding the biological activity of the subjects' wild-type and mutant recombinant eel LH and eel FSH. These results are discussed in the later part of the manuscript, and according to the author's suggestion support their findings that the EC50 and the maximal responsiveness of the mutants were increased observed for both the wild-type gonadotropins. However, in my opinion, to formulate such a statement, it is necessary to conduct a statistical analysis that will confirm the observations made by the authors. Simply comparing of average values of cAMP levels and Rmax average cAMP levels does not allow them to make such conclusions.
→We do not normally statistical analysis in the cAMP EC50 value.
To summarize, due to the interesting topic and the authors' effort put into the experimental part of the work, I believe that the manuscript can be accepted for publication after a major revision.

Round 2
Reviewer 2 Report
Comments and Suggestions for Authors
The manuscript has been well improved following the criticisms of the three reviewers. Nevertheless, it could be further improved.
It should be stated in the title that the recombinant hormones under study are all single-chain analogs.
Previous papers concerning beta-alpha single-chain eLH/CG (thus, with the eLH/CG CTP as a linker) are not mentioned.
Wouldn't it be possible to use cells transfected with the eel LH or FSH receptor to assess the binding and in vitro bioactivities of the different hormones in a homologous system to support the contention that these hormones would be of interest in the control of reproduction in eels ?
Author Response
Comments and Suggestions for Authors
The manuscript has been well improved following the criticisms of the three reviewers. Nevertheless, it could be further improved.
It should be stated in the title that the recombinant hormones under study are all single-chain analogs.
→We changed and inserted “single-chain” in the title.
Previous papers concerning beta-alpha single-chain eLH/CG (thus, with the eLH/CG CTP as a linker) are not mentioned.
→We inserted “as a linker” in the line 68-69 as reviewer’s comment.
Wouldn't it be possible to use cells transfected with the eel LH or FSH receptor to assess the binding and in vitro bioactivities of the different hormones in a homologous system to support the contention that these hormones would be of interest in the control of reproduction in eels?
→Of course, we tested eel oocytes with rec-eel LH and FSH hormones in vitro in the previous studies. But we think that the transfected cells with those receptors are good system of the biological activity analysis. And we are engaging in the ovulation induction studies by the injection to eel. Although we did not show the results in the presented studies, we are going to report a good effect of the results in vivo.